# Descriptive Comparative Transcriptomic Analysis of Genotype IV SHEV ORF3-Expressing HepG2 Cells

**DOI:** 10.3390/microorganisms13020412

**Published:** 2025-02-13

**Authors:** Hanwei Jiao, Chi Meng, Fengyuan Jiao, Gengxu Zhou, Lingjie Wang, Shengping Wu, Cailiang Fan, Jixiang Li, Liting Cao, Yu Zhao, Yichen Luo

**Affiliations:** 1The College of Veterinary Medicine, Southwest University, Chongqing 402460, China; jiaohanwei@swu.edu.cn (H.J.); mengchi@email.swu.edu.cn (C.M.); senrendipity@email.swu.edu.cn (F.J.); zgx973589243@email.swu.edu.cn (G.Z.); guolicheng666@email.swu.edu.cn (L.W.); chemie@email.swu.edu.cn (S.W.); 13308380005@163.com (C.F.); swu_lucky@163.com (J.L.); caoliting@swu.edu.cn (L.C.); 2Animal Epidemic Prevention and Control Center of Rongchang, Chongqing 402460, China; 3Institute of Animal Husbandry and Veterinary Medicine of Guizhou Academy of Agricultural Science, Ministry of Agriculture and Rural Affairs Key Laboratory of Crop Genitic Resources and Germplasm Innovation in Karst Region, Guiyang 550005, China

**Keywords:** SHEV, ORF3, HepG2, transcriptome, mRNA

## Abstract

**Background:** Swine hepatitis E (HEV) is a zoonotic infectious disease caused by the swine hepatitis E virus (SHEV). Open reading frame 3 (ORF3) is a key virulence factor in swine HEV, playing a crucial role in the release of viral particles, the modulation of the host innate immune response, and regulation of autophagy and apoptosis, etc. However, its main function and pathogenic mechanism remain incompletely understood. **Results:** In our study, adenoviruses ADV4-ORF3 and ADV4-GFP were successfully constructed and mediated the overexpression of enhanced green fluorescent protein (EGFP)-ORF3 and EGFP in HepG2 cells. A total of 217 differentially expressed messenger RNAs (mRNAs) were screened by high-throughput sequencing, and 27 statistically significant differentially expressed genes were screened for further quantitative real-time reverse transcription (qRT-PCR) verification by functional enrichment (Gene Ontology [GO] and Kyoto Encyclopedia of Genes and Genomes [KEGG]). They are mainly involved in six pathways: the cellular response to unfolded protein, inflammatory response, cytokine activity, TNF signaling pathway, influenza A, and pathways in cancer. In a comparative analysis of transcriptome and mRNA expression profiles of lncRNA sequencing, the results showed that 3 mRNAs of GPX1, MDM4, and CLDN and 39 transcripts overlapped and have been identified. **Conclusions:** Eight differential genes, HSPA1A, HSPA1B, PLD3, RELA, GPI, SAMHD1, RPS6KA4, and PIK3CB, were successfully verified. Comparing and analyzing the results of the two sequencing methods indicated that the 3 mRNAs of GPX1, MDM4, and CLDN and 39 transcripts overlapped and have been identified in SHEV ORF3-expressing HepG2 cells, which has laid a genetic foundation for the physiological function and mechanism of SHEV ORF3.

## 1. Introduction

Hepatitis E virus (HEV) is a single-stranded, positive-sense RNA virus that belongs to the hepeviridae family. The majority of concerning HEV genotypes belong to the paslahepevirus genus and are subsequently divided into eight genotypes [1]. Genotype IV SHEV, isolated from piglets in the United States, is a prominent zoonotic strain [2,3,4]. Although HEV in pigs only causes mild lesions of hepatitis without clinical symptoms, there is a risk of zoonotic infection in humans through direct contact with infected pigs or the consumption of undercooked pork [5,6,7].

HEV belongs to the hepatitis virus family. So far, at least four major genotypes have been found in mammals: genotype 1 and genotype 2 only have human hosts, while genotype 3 and genotype 4 are zoonotic. Therefore, all HEV strains isolated from pigs in the world currently belong to genotype 3 or 4. HEV is a spherical virus particle without an envelop, with a diameter of 27–34 nm [8,9,10,11]. SHEV cannot effectively proliferate on cells. The genotype of SHEV is a 7.2 kb polyadenylation, single-stranded positive-stranded RNA molecule. Its genome contains three open reading frames (ORFs), a short 5′ non-coding region (NCR), and a short 3′ NCR. ORF1 encodes a non-structural protein, ORF2 encodes an immunogenic capsid protein, and ORF3 encodes a small multifunctional protein. ORF2 and ORF3 are translated from a polycistronic mRNA, and the two parts overlap with each other but do not overlap with ORF1 [12,13].

The transmission route of HEV among pigs may be fecal–oral transmission, and infected pig feces may be the main source of virus transmission. It is speculated that the infection routes of pigs include direct contact with infected pigs or the consumption of feed or water sources contaminated with feces. Therefore, replicating HEV infection mainly occurs through oral routes [14]. However, other transmission routes cannot be ruled out [14,15,16]. By appropriate cooking, the infectious activity of SHEV in commercial pig liver can be completely inactivated, such as frying or boiling for more than 5 min. Heating at 56 °C for a long time (1 h) or at a higher temperature (>65 °C) for a short time can successfully inactivate the HEV virus. SHEV can maintain activity in acidic and weakly alkaline environments within the intestine [17,18,19].

The pathogenic mechanism of SHEV is still unclear. Scientists believe that SHEV replicates in the gastrointestinal tract after ingestion and then spreads to its target organ, the liver. The replication of viruses in the liver has been proven. SHEV has also been isolated from various organs outside the liver, including the small intestine, colon, liver, and mesenteric lymph nodes [20,21]. SHEV ORF3 is a regulatory protein that alters the activity of selected transcription factors and cytoplasmic signaling pathways [22]. The ORF3 protein interacts with microtubules and various host cell proteins to regulate the host environment for HEV replication. Importantly, the ORF3 protein can promote the budding of viral particles into multivesicular bodies. Finally, the polycystic body fuses with the plasma membrane, and viral particles are released from liver cells into the bloodstream or degraded by bile acid salts in the bile duct [23].

In this study, we overexpressed the genotype IV SHEV ORF3 protein in HepG2 and analyzed the transcriptome data by transcriptome sequencing. To be more precise, we explore how SHEV ORF3 affects mRNAs in host cells; at the same time, we integrate multiple omics results for a comparative analysis to reveal the impact of SHEV ORF3 on transcriptome changes in target cells, laying a foundation for revealing the pathogenic mechanism of SHEV.

## 2. Materials and Methods

### 2.1. Cell Lines

HepG2, 293A, and 293T cells were purchased from the Shanghai Cell Bank, Chinese Academy of Sciences, and cultured in Dulbecco’s modified Eagle medium (DMEM) containing 10% fetal bovine serum, 10% penicillin, and streptomycin at 37 °C, 5% CO_2_ (Life Technology, Carlsbad, CA, USA).

### 2.2. Preparation of Recombinant Adenovirus AD-ORF3 and AD-GFP

The recombinant adenovirus AD-ORF3 and AD-GFP were generated by Shanghai GenePharma Co, Ltd. (Shanghai, China) [24].

As we previously described [25], HepG2 cells were infected with a multiplicity of infection (MOI) of 5:1; the HepG2 cell samples infected with adenovirus ADV4-ORF3 were named AD-ORF3, and the HepG2 cell samples infected with adenovirus ADV4-GFP were named AD-GFP, and they were the control group.

### 2.3. RNA Sample Preparation and Transcriptome Sequencing

RNA samples Ad_ORF3_3, Ad_ORF3_2, Ad_ORF3_1, Ad_GFP_3, Ad_GFP_2, and Ad_GFP_1 underwent transcriptome sequencing (LC-Bio Technology CO., Ltd., Hangzhou, China) [26].

### 2.4. Bio-Informatics Analysis

The cutadapt filter of the clean reads was aligned with the human genome reference using HISAT 2, and the expression levels of the transcripts were assembled and estimated using Stringtie software (v2.2.3). DESeq 2 software was used to analyze the differential expression of genes between two different groups, and then the differentially expressed genes were obtained. Gene Ontology (GO) and the Kyoto Encyclopedia of Genes and Genomes (KEGG) were used to analyze the functional enrichment of differentially expressed genes, and the enrichment results were displayed in the form of bubble plots.

### 2.5. qRT-PCR Validation

Quantitative RT-PCR (qRT-PCR) was used to quantitatively detect the mRNA level of differentially expressed genes. Approximately 1 μg of total RNA samples was reverse-transcribed using the PrimeScript™ RT kit. Subsequently, qRT-PCR validation was performed using TB Green^®^ Premix Ex Taq™ II (Tli RNaseH Plus) (Takara, Dalian, China). The β-actin gene was used as the internal reference gene and was analyzed by the 2^−ΔΔCt^ method. Three biological replicates were set up for qRT-PCR analysis. The data processing was carried out using Graph-Prism 8 software, and it was considered that *p* < 0.05 was statistically significant and significantly different, which was represented by an asterisk, and the difference of *p* < 0.01 was extremely significant, which was represented by two asterisks. We used NCBI online software (https://www.ncbi.nlm.nih.gov/tools/primer-blast) for the primer design. Genes and primers are listed in Table 1.

### 2.6. Comparative Analysis of Multiple Omics

HepG2 cells were infected with the recombinant adenovirus of ADV4-NC (control) and ADV4-ORF3 for 24 h, total RNA and depleted ribosomal RNA were extracted, and paired-end lncRNA sequencing was performed. We selected significantly differentially expressed genes (│Log_2_FC│ ≥ 1 and q ≤ 0.5) from the transcriptome and mRNA expression profiles of the lncRNA sequencing results. A Venn diagram reference text advanced pie plot was performed using the OmicStudio tools at https://www.omicstudio.cn/tool (accessed on 6 January 2025).

## 3. Results

### 3.1. Transcriptome Sequencing Results

#### 3.1.1. Overview of Transcriptome

Cutadapt was used to filter out out-of-spec sequences to obtain valid data, which were then aligned to a reference genome, and finally the transcript assembly and quantification were performed. A total of 44.2 G of raw data were obtained from the RNA sequencing, of which 43.4 G were effective data, accounting for more than 98% of the effective data, and the Q20 of all samples reached 99.98% (Table 2). There is a high correlation between samples (Figure 1).

According to the reference genome information, the valid data were compared according to exons, introns, and intergenic, and the results showed that the proportion of exons in each group of samples was more than 97%, the proportion of introns was more than 2%, and the rest was the proportion of intergenic, which met the requirements of the sequencing and analysis (Figure 2). In addition, the distribution of gene expression values in the six samples is shown in a box plot. In the box plot, the overall level of each sample is basically the same, and it is reproducible (Figure 3).

#### 3.1.2. Differential Genes and Transcripts

A total of 217 differentially expressed genes (1635 transcripts) were obtained, including 86 up-regulated genes and 131 down-regulated genes, compared with Ad-ORF3 and Ad-GFP (Figure 4). According to the similarity of the gene expression profile of the sample, the genes were clustered and analyzed, and the expression of genes in the samples is visually displayed (Figure 5). Heat maps of differentially expressed genes and transcripts are shown in Figure 6. GO and KEGG enrichment analyses were performed on the differential genes. The distribution of the number of differentially differentiated genes on the GO item enriched by biological processes, cell components, and molecular functions is displayed in the form of histograms, and the top 20 results of the GO and KEGG enrichment analysis are further displayed in the form of scatter plots through ggplot2 (Figure 7).

The GO analysis showed that the GO terms of differentially expressed mRNA were as follows: significantly enriched C3HC4−type RING finger domain binding (GO:0055131), cellular response to unfolded protein (GO:0034620), negative regulation of inclusion body assembly (GO:0090084), protein folding chaperone (GO:0044183), chaperone cofactor-dependent protein refolding (GO:0051085), inflammatory response (GO:0006954), cytokine activity (GO:0005125), mesodermal cell fate determination (GO:0007500), positive regulation of nucleotide-binding oligomerization domain containing 2 signaling pathways (GO:0070434), glomerular capillary formation (GO:0072104), type I interferon signaling pathway (GO:0060337), fat cell differentiation (GO:0045444), negative regulation of cell population proliferation (GO:0008285), negative regulation of collagen biosynthetic process (GO:0032966), regulation of cellular response to heat (GO:1900034), protein refolding (GO:0042026), neutrophil chemotaxis (GO:0030593), cellular heat acclimation (GO:0070370), negative regulation of transcription by RNA polymerase II (GO:0000122), and negative regulation of cysteine-type endopeptidase activity involved in apoptotic process (GO:0043154) (Figure 8).

The KEGG analysis showed that the pathways of the following differentially expressed mRNAs were significantly enriched: Legionellosis (ko05134), TNF signaling pathway (ko04668), cytokine−cytokine receptor interaction (ko04060), toxoplasmosis (ko05145), influenza A (ko05164), pathways in cancer (ko05200), herpes simplex virus 1 infection (ko05168), leukocyte transendothelial migration (ko04670), prion diseases (ko05020), NOD-like receptor signaling pathway (ko04621), chemical carcinogenesis (ko05204), rheumatoid arthritis (ko05323), IL-17 signaling pathway (ko04657), steroid hormone biosynthesis (ko00140), protein processing in endoplasmic reticulum (ko04141), longevity regulating pathway-multiple species (ko04213), basal cell carcinoma (ko05217), epithelial cell signaling in Helicobacter pylori infection (ko05120), antigen processing and presentation (ko04612) and chemokine signaling pathway (ko04062) (Figure 9).

### 3.2. Further Screening of ORF3 Involvement in the Pathway

Further analysis of the GO enrichment and KEGG pathways of differentially expressed mRNAs revealed that six pathways with a high correlation of ORF3 were obtained, which were cellular response to unfolded protein (GO:0034620), inflammatory response (GO:0006954), cytokine activity (GO:0005125), TNF signaling pathway (ko04668), influenza A (ko05164), and pathways in cancer (ko05200). At the same time, according to the significance column of these six pathways, │Log_2_FC│ ≥ 1 and q ≤ 0.5 were selected as the conditions, the first five differentially expressed genes were selected to draw heat maps, and primers were designed for qRT-PCR verification. A total of 27 differential genes were screened, of which 15 genes were up-regulated and 12 genes were down-regulated (Figure 10).

### 3.3. qRT-PCR

In order to further confirm the sequencing results, qRT-PCR verification was performed on the 27 differentially expressed genes. But only ENST00000375651-HSPA1A, ENST00000375650-HSPA1B, ENST00000409419-PLD3, ENST00000308639-RELA, ENST00000356487-GPI, ENST00000647095-SAMHD1, ENST00000528355-RPS6KA4, and ENST00000477593-PIK3CB were successfully verified, and the expression of these eight differential genes was consistent with the transcriptome sequencing. Genes ENST00000308639-RELA, ENST00000409419-PLD3, ENST00000356487-GPI, ENST00000528355-RPS6KA4, and ENST00000477593-PIK3CB were down-regulated compared with the control group, and genes ENST00000375651-HSPA1A, ENST00000375650-HSPA1B, and ENST00000308639-RELA were up-regulated compared with the control group, and there were significant differences (Figure 11).

### 3.4. Comparative Analysis of Transcriptome and mRNA Expression Profiles of lncRNA Sequencing

Our previous research found that 319 significantly differentially expressed lncRNAs were identified. At the same time, we also identified significantly differentially expressed mRNA profiles, including 62 genes and 6564 transcripts (│Log_2_FC│ ≥ 1 and q ≤ 0.5) [25].

In this study, a notable 217 differentially expressed mRNAs were identified through high-throughput sequencing. Twenty-seven of these genes were validated through qRT-PCR, with significant involvement in pathways related to unfolded protein response, inflammatory response, cytokine activity, and others.

Finally, we performed a Venn analysis by taking the intersection of the 217 mRNAs from the transcriptomics and 62 mRNAs from the lncRNA sequencing results. The results showed that three mRNAs were the same; they were GPX1, MDM4, and CLDN10 (Figure 12A). And we performed a Venn analysis on 1635 significantly differentially expressed transcripts from the transcriptome sequencing and 6564 significantly differentially expressed transcripts from the lncRNA sequencing. The results showed that 39 transcripts were identified (Figure 12B); they were ENST00000404814, ENST00000639300, ENST00000381486, ENST00000530741, ENST00000426481, ENST00000533455, ENST00000541297, ENST00000268676, ENST00000472765, ENST00000394126, ENST00000419510, ENST00000489337, ENST00000261168, ENST00000497866, ENST00000651608, ENST00000356271, ENST00000433206, ENST00000375651, ENST00000567695, ENST00000261963, ENST00000647211, ENST00000529301, ENST00000351328, ENST00000534064, ENST00000373997, ENST00000361571, ENST00000644483, ENST00000474967, ENST00000616406, ENST00000553474, ENST00000493560, ENST00000550785, ENST00000409155, ENST00000490898, ENST00000413364, ENST00000452898, ENST00000392379, ENST00000448507, and ENST00000532636. Comparing the results of the two sequencing methods, it was found that 3 mRNAs and 39 transcripts overlapped and have been identified in SHEV ORF3- expressing HepG2 cells.

## 4. Discussion

Hepatitis E virus (HEV) is one of the main causes of acute hepatitis in humans due to fecal–oral transmission, and it has the potential for cross-species transmission. In addition to humans, pigs are a common reservoir for HEV [27,28]. ORF3 is involved in the release of HEV particles, and it is involved in a variety of signaling pathways to regulate host cell innate immunity. Our study compared the transcriptomes of HepG2 cells overexpressing ORF3 and GFP at a high level, conducted a comprehensive analysis of differentially expressed mRNAs, and identified a total of 217 differentially expressed genes, including 86 up-regulated genes and 121 down-regulated genes, and summarized their general characteristics and functional annotations, which provided new ideas for the study of ORF3.

HEV ORF3 is involved in a variety of signaling pathways, inhibits host immune responses, and increases its own survival. Interferon-induced antiviral response plays an important role in the innate immune response, and HEV ORF3 interferes with the innate immune response by down-regulating TLR3 and TLR7, impairing the production of host endogenous type I interferon, while also inhibiting the activation of the NF-κB, JAK/STAT, and JNK/MAPK signaling pathways [29,30]. GO and KEGG pathway enrichment analyses revealed the substantial involvement of six key pathways, including unfolded protein response, cytokine activity, and TNF signaling, so that many differentially expressed genes were annotated to biological functions, and six pathways with ORF3 were highly involved, namely cellular response to unfolded protein (GO:0034620), inflammatory response (GO:0006954), cytokine activity (GO:0005125), TNF signaling pathway (ko04668), influenza A (ko05164), and pathways in cancer (ko05200).

It has also been shown that genotype 1 HEV participates in unfolded protein response (UPR) and transiently activates NF-κB signaling through the ORF3 activation of transcription factor 6 (ATF6) in the early stage, and it inhibits TNF-α-induced NF-κB signaling in the later stage, which can provide a favorable environment for viral replication [31]. In addition, HEV ORF3 can inhibit the inflammatory response of macrophages to lipopolysaccharide (LPS), and damage the phagocytosis of macrophages by down-regulating the expression of CD14 and CD64 [32].

In this study, the ENST00000375651-HSPA1A, ENST00000375650-HSPA1B, ENST00000409419-PLD3, ENST00000308639-RELA, ENST00000356487-GPI, ENST00000647095-SAMHD1, ENST00000528355-RPS6KA4, and ENST00000477593-PIK3CB differential genes were further screened and successfully verified, and these eight differential genes were mainly involved in the cellular response to unfolded protein (GO:0034620); HEV ORF3 regulated A20 primarily via activating transcription factor 6 (ATF6), involved in unfolded protein response (UPR) [33]; through inflammatory response (GO:0006954), HEV-4 placental infection resulted in serious histopathological damage, such as fibrosis and calcification, and severe inflammatory responses [26]; in terms of cytokine activity (GO:0005125), a comprehensive screening of human cytokines and chemokines revealed that IFN-alpha was the sole humoral factor inhibiting HEV replication [34]; in the TNF signaling pathway (ko04668), a significantly higher level of TNF-alpha was observed in HEV-infected pregnant women than non-HEV pregnant women [35]; influenza A (ko05164), HEV, and the influenza A virus (IAV) are two important pathogens that can infect humans and various animals causing public health problems [36].

In addition, in our previous study, by comparing the expression profile and structure of differentially expressed lncRNA cis target genes and mRNAs in HepG2 cells overexpressing swine HEV ORF3, it was revealed that ORF3 may affect the interconversion of pentose and glucuronic acid and mediate the formation of obstructive jaundice by affecting bile secretion, which is of great significance for the function and pathogenesis of HEV ORF3 [24]. Transcriptomic sequencing was performed using the polyA enrichment method and 217significantly differentially expressed mRNAs were identified, while only 62 significantly differentially expressed mRNAs were identified through lncRNA sequencing using the ribosome removal method, resulting in differences between the two sequencing results. Meanwhile, lncRNA sequencing mainly focuses on exploring differentially expressed lncRNAs, which may not be sufficient for identifying the abundance of differentially expressed mRNAs. Now, we have further analyzed the transcriptome and supplemented and improved the function and pathogenesis of ORF3 at the gene level, laying the genetic foundation of HEV ORF3.

## 5. Conclusions

Our study revealed that HEV infection can lead to differentially expressed HSPA1A, HSPA1B, PLD3, RELA, GPI, SAMHD1, RPS6KA4, and PIK3CB after HEV infection; the differential genes HSPA1A, HSPA1B, PLD3, RELA, GPI, SAMHD1, RPS6KA4, and PIK3CB, were successfully verified. Comparing and analyzing the results of the two sequencing methods indicated that 3 mRNAs of GPX1, MDM4, and CLDN and 39 transcripts overlapped and have been identified in SHEV ORF3-expressing HepG2 cells, which has laid a genetic foundation for the physiological function and mechanism of SHEV ORF3.

## Figures and Tables

**Figure 1 microorganisms-13-00412-f001:**
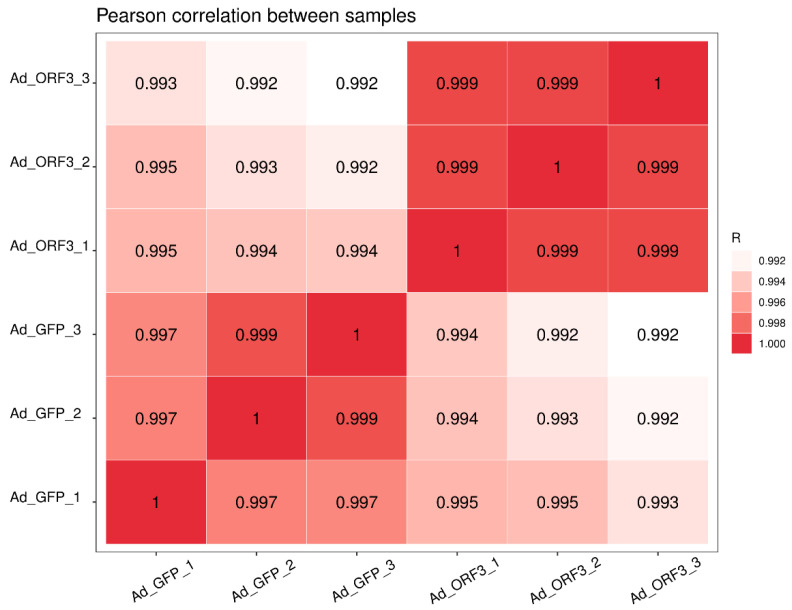
Pearson correlation between samples. The horizontal and vertical coordinates represent the samples, and the color shading indicates the magnitude of the correlation coefficient between the two samples. The closeness to red (the closer the coefficient is to 1) indicates a greater correlation. The closer to white, the less the correlation.

**Figure 2 microorganisms-13-00412-f002:**
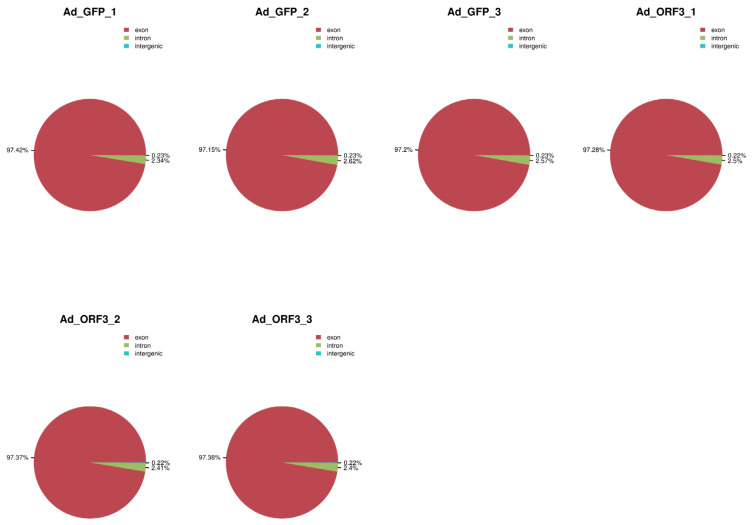
Alignment of gene distribution with reference genome. The red part represents the exons, the green part represents the introns, and the blue part represents the intergenic, and the percentage content of sequencing sequence localization in the exon region and the intron region should be the highest. Indicates that the requirements for sequencing and analysis are met.

**Figure 3 microorganisms-13-00412-f003:**
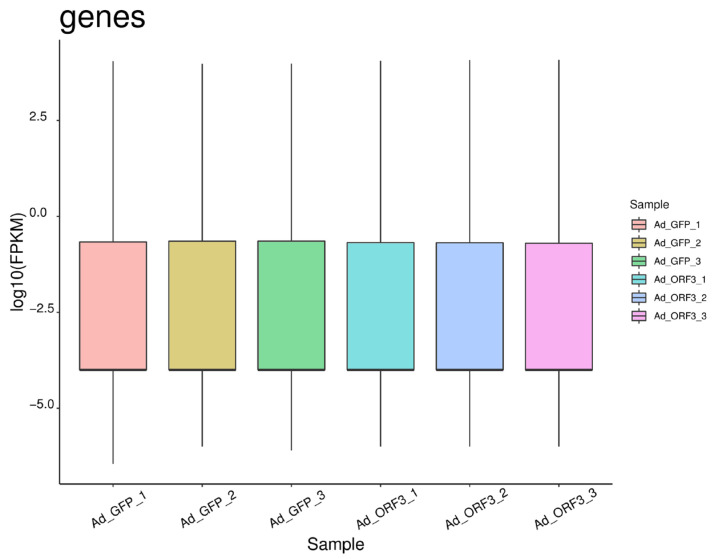
Distribution of gene expression values across 6 samples. The distribution statistics of gene expression values in the sample are represented by FPKM box plots, which can illustrate the expression level of genes from the overall level.

**Figure 4 microorganisms-13-00412-f004:**
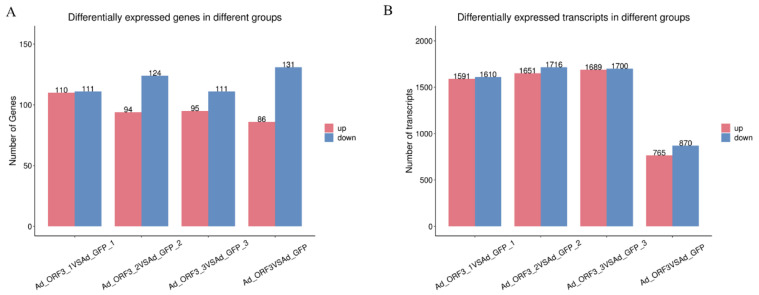
Differentially expressed genes (**A**) and transcripts (**B**) in different groups. Differentially expressed genes are plotted into histograms to visually indicate the expression of differentially expressed genes within a group, where red represents gene up-regulation and blue represents gene down-regulation.

**Figure 5 microorganisms-13-00412-f005:**
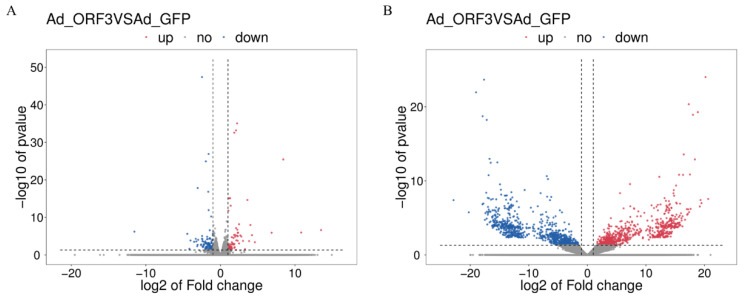
Volcano plots of differentially expressed genes (**A**) and transcripts (**B**) in Ad-ORF3 and Ad-GFP samples. The abscissa represents the differential expression fold of the gene in different samples. The ordinate represents the statistical significance of the difference in gene expression levels. Red represents up-regulated significantly differentially expressed genes, blue represents down-regulated significantly differentially expressed genes, and gray represents non-significantly differentially expressed genes.

**Figure 6 microorganisms-13-00412-f006:**
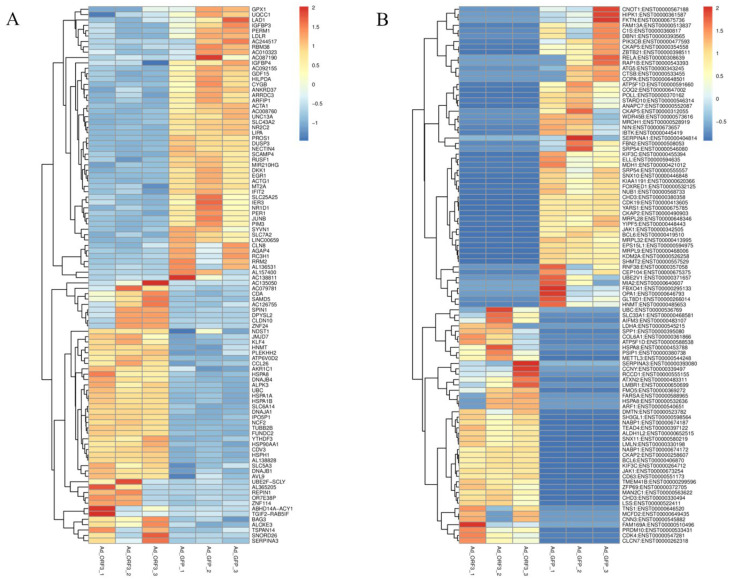
Heat maps of differentially expressed genes (**A**) and transcripts (**B**). The abscissa represents the sample, the ordinate is the differential gene or transcript, the different colors in the figure represent the different levels of gene expression, the color from blue to white to red represents the expression level from low to high, red represents highly expressed genes, and blue represents low-expression genes.

**Figure 7 microorganisms-13-00412-f007:**
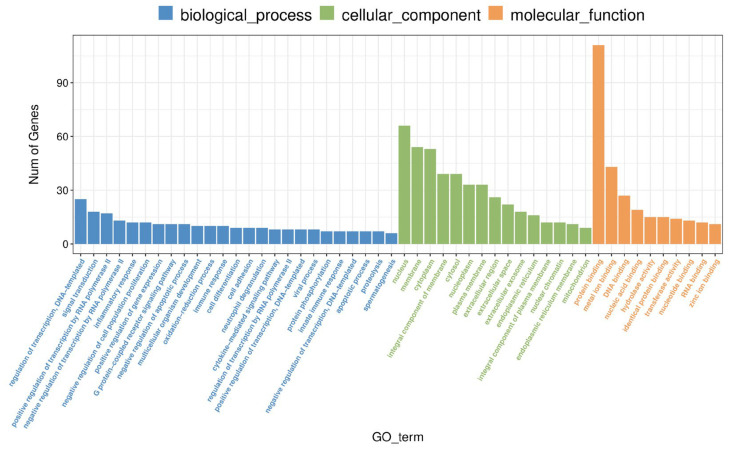
Histogram of GO enrichment pathway of differential genes. The GO enrichment classification histogram reflects the distribution of the number of distinct genes on the enrichment entries in biological processes, cellular components, and molecular processes, and due to the large number of enriched GO terms, it is impossible to put all the results in one graph, so the top 25, top 15, and top 5 of BP, CC, and MF are displayed respectively for plotting, where the abscissa represents GO terms and the ordinate represents the number of genes.

**Figure 8 microorganisms-13-00412-f008:**
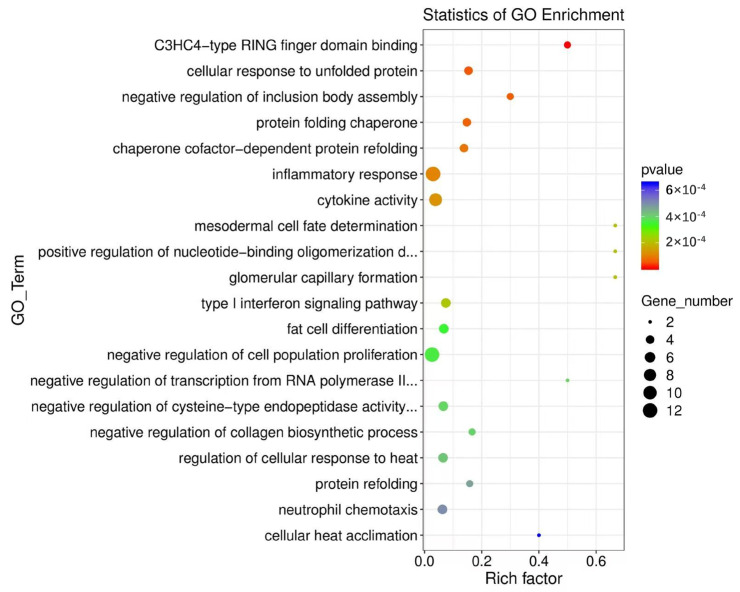
GO functional enrichment analysis of differentially expressed mRNA genes. The results of the GO enrichment analysis are displayed in the form of a bubble diagram, in which the abscissa rich factor represents the proportion of the number of differential genes located in the GO to the total number of genes located in the GO (the larger the rich factor, the higher the degree of GO enrichment), the ordinate is the GO term, the size of the bubble represents the gene number, and the bubble color represents the *p*-value of the enrichment analysis.

**Figure 9 microorganisms-13-00412-f009:**
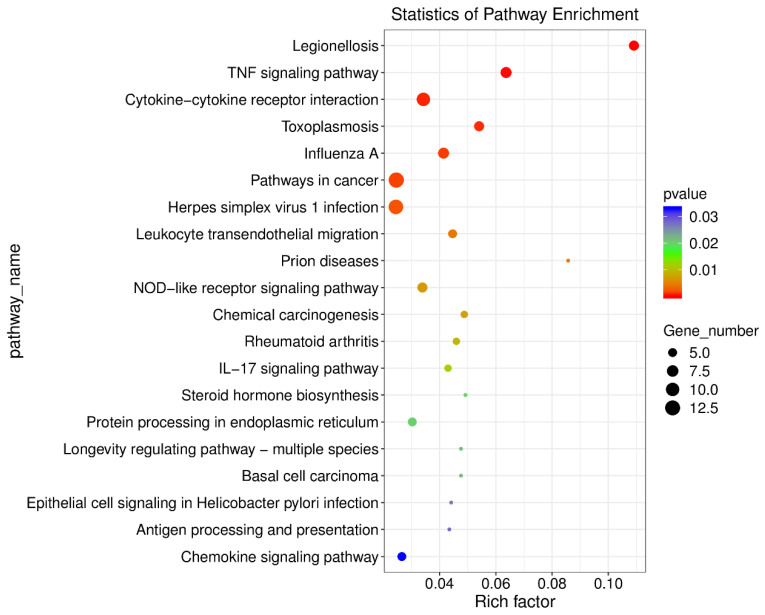
KEGG pathway enrichment analysis of differentially expressed mRNA genes. The results of the KEGG enrichment analysis are displayed in the form of bubble diagrams, in which the abscissa represents the proportion of the number of differential genes located in the pathway to the total number of genes located in the pathway, and the larger the rich factor, the higher the degree of pathway enrichment. The ordinate is the KEGG pathway, and in the bubble diagram, the size of the bubble represents the gene number, and the color of the bubble represents the enrichment’s significance.

**Figure 10 microorganisms-13-00412-f010:**
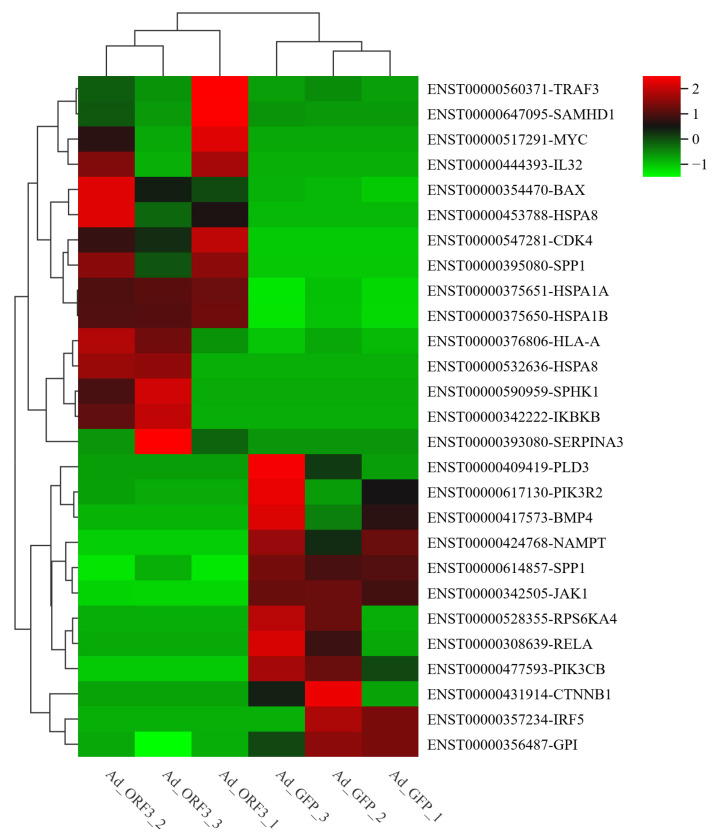
The differentially related genes highly correlated with ORF3 were further obtained by KEGG and GO enrichment analysis. The heat map shows 27 differentially expressed genes, with different colors representing different expression levels, red indicating high-expression genes and green indicating low-expression genes.

**Figure 11 microorganisms-13-00412-f011:**
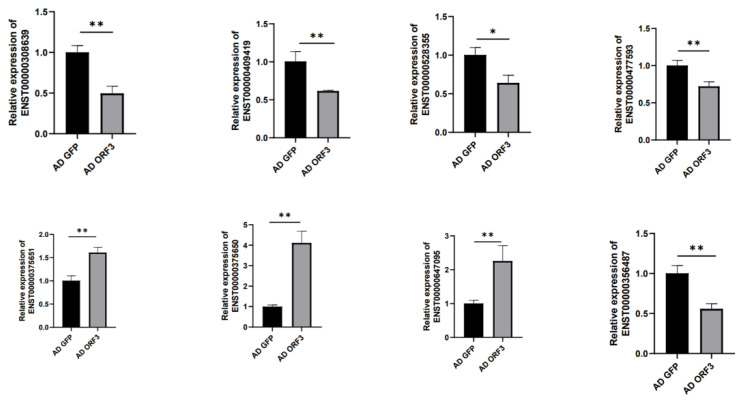
Eight differential genes were validated by qRT-PCR. “*”indicates significant difference and “* *” indicates extremely significant difference.

**Figure 12 microorganisms-13-00412-f012:**
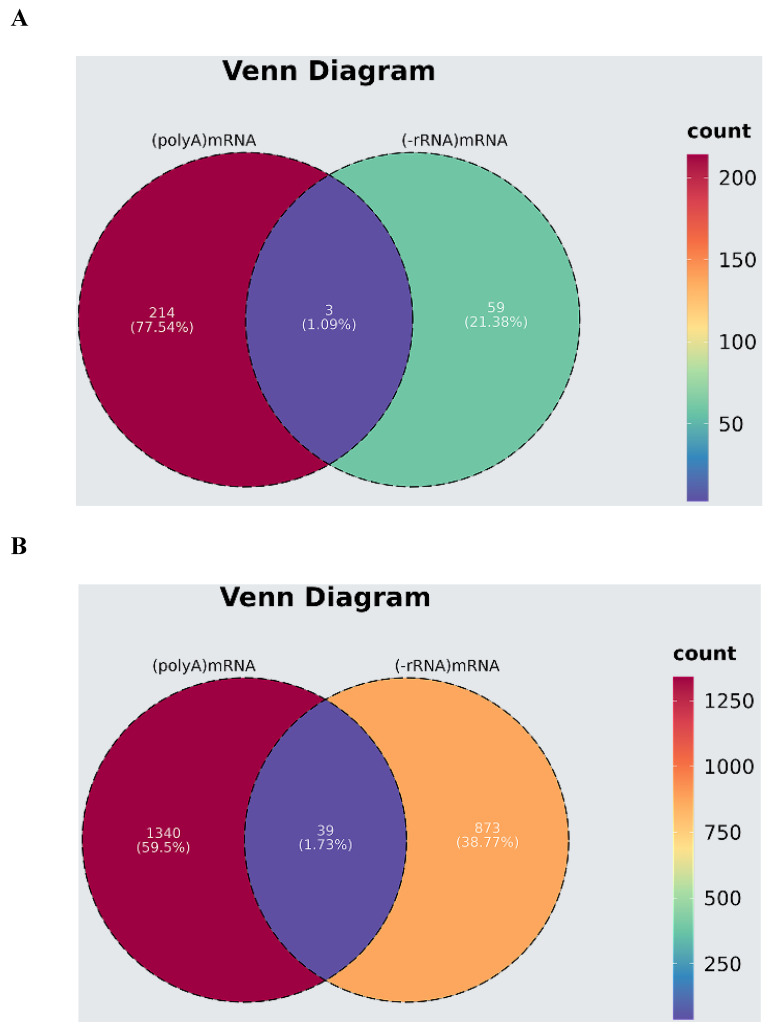
Comparing transcriptome and mRNA expression profiles of lncRNA sequencing results through Venn analysis. (**A**) Venn analysis for 217 significantly differentially expressed mRNAs from transcriptomics and 62 mRNAs from lncRNA sequencing results. (**B**) Venn analysis for 1635 significantly differentially expressed transcripts from transcriptome sequencing and 6564 transcripts of mRNAs from lncRNA sequencing.

**Table 1 microorganisms-13-00412-t001:** Primers for selected significantly differentially expressed RNAs (*p* ≤ 0.05 and q ≤ 1) for qRT-PCR validation.

Gene Name	Transcripts Name	Primer Sequence (5′-3′)
HSPA8	ENST00000532636	F: TCTTTCACCCAGTTCCCTGC
		R: TGGACATGGTTGCCCCATAC
HSPA8	ENST00000453788	F: TTGGGTCTTGTAAGGGCAGC
		R: GACATGGTTGCTGGGGTGTA
BAX	ENST00000354470	F: CCCTTTTGCTTCAGGGGATGA
		R: GAAGTCCAATGTCCAGCCCA
HSPA1A	ENST00000375651	F: ATAAAAGCCCAGGGGCAAGC
		R: AACACTGGATCCGCGAGAAG
HSPA1B	ENST00000375650	F: GCAGGTGTGTAACCCCATCA
		R: GAGTCCCAACAGTCCACCTC
SPP1	ENST00000395080	F: AGGCATCACCTGTGCCATAC
		R: GGCCACAGCATCTGGGTATT
SERPINA3	ENST00000393080	F: TACATCCAGCTCCCTGAGAGT
		R: TCAGGGGCCTTCAGGACTAA
SPHK1	ENST00000590959	F: CAGCACCGATAAGGAGCTGAA
		R: AACCAAAGAGGCCACGTCC
PLD3	ENST00000409419	F: TGCCCACTTTTGTTCTGCCA
		R: CCTTCCACGCCTCAATCTCA
RELA	ENST00000308639	F: GCTCGTCTGTAGTGCACGC
		R: TGATCTCCACATAGGGGCCA
IL32	ENST00000444393	F: ACCAGCTGAGTATTTGTGCCA
		R: CGGCCAAAAGTTCAAGGAGC
GPI	ENST00000356487	F: GTGTACCTTCTAGTCCCGCC
		R: ATGCCCATGGTTGGTGTTGA
SPP1	ENST00000614857	F: AATTCTGGGAGGGCTTGGTT
		R: TTTTCCTTGGTCGGCGTTTG
BMP4	ENST00000417573	F: AAGCCGAGGCGAGAGAGAC
		R: CTCGGGATGGCACTACGGAA
NAMPT	ENST00000424768	F: CCATGCCTCCAGTTTCGAGAT
		R: TATCTGGGGGCAGTGATGGT
HLA-A	ENST00000376806	F: CGGTGTATGGATTGGGGAGT
		R: GCTTCTCTGGAAACCCGACA
SAMHD1	ENST00000647095	F: TGACATGTTCCACACTCGCA
		R: ACTCTAGCCAAGTATCAAGGAAAA
JAK1	ENST00000342505	F: ATCCTGGAGCTGCAGACAGT
		R: AGCTACTTCAGAGAAGCGCA
IRF5	ENST00000357234	F: CAGAGCTCAGCTTGGTCCC
		R: GGATGGACTGGTTCATGGCA
IKBKB	ENST00000342222	F: CGTGTCCTTCAGGGAGAGTG
		R: TCCCCACGAATGATGTGCAA
PIK3R2	ENST00000617130	F: AGGCCATTGAAAGGACAGGG
		R: GTGCCAGCAGGAAGCTCTTA
RPS6KA4	ENST00000528355	F: GAGGACGACGATGAGAGCTG
		R: CTTGCCGTAGGCTTCTCCTC
PIK3CB	ENST00000477593	F: TTGGAATAGTAGCAGGCGGC
		R: CGCCCAGATGTCAAGGATGT
TRAF3	ENST00000560371	F: ACGGACCGCGAGAACTCC
		R: CTTTAGCGGCGGGTTAGTCT
CDK4	ENST00000547281	F: GCTGGCGTGAGTGTACAAGG
		R: CAGTCGCCTCAGTAAAGCCA
MYC	ENST00000517291	F: CAAAAGAAAATGCAGCGGGC
		R: CTAACGTTGAGGGGCATCGT
CTNNB1	ENST00000431914	F: CCCGGTGATTCAGGTCGAAA
		R: GAGTAGCCATTGTCCACGCT

**Table 2 microorganisms-13-00412-t002:** Sample quality control statistical table.

Sample	Raw Data	Valid Data			Valid Ratio (Reads)	Q20%	Q30%	GC Content %
	Read	Base	Read	Base				
Ad_GFP_1	54,701,040	8.21 G	53,701,618	8.06 G	98.17	99.98	98.23	49
Ad_GFP_2	49,681,178	7.45 G	48,784,570	7.32 G	98.20	99.98	98.48	49
Ad_GFP_3	49,987,694	7.50 G	49,061,438	7.36 G	98.15	99.98	98.37	49
Ad_ORF3_1	51,214,508	7.68 G	50,288,760	7.54 G	98.19	99.98	98.36	49
Ad_ORF3_2	47,990,420	7.20 G	47,125,072	7.07 G	98.20	99.98	98.34	49
Ad_ORF3_3	41,049,826	6.16 G	40,340,186	6.05 G	98.27	99.98	98.43	49

## Data Availability

The original contributions presented in the study are included in the article, further inquiries can be directed to the corresponding author.

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
