# Peer review of "Descriptive Comparative Transcriptomic Analysis of Genotype IV SHEV ORF3-Expressing HepG2 Cells"

_microorganisms, 2025, doi:10.3390/microorganisms13020412_

Round 1

Reviewer 1 Report

Comments and Suggestions for Authors

Jiao et al focused on the role of the ORF3 protein of the Hepatitis E virus (HEV) in infected cells. To this end, a transcriptome analysis of HepG2 cells transduced with adenovirus strains expressing the green fluorescent protein (GFP) alone or GFP tagged-ORF3 was performed.

The first part of the article describes the validation of the transcriptome sequencing method. The quality scores and sequence recovery was sufficiently high to compare the different samples and perform a comparative analysis between cells expressing GFP and GFP-ORF3.

The authors found 217 differentially expressed genes (and 1635 transcripts) and used Gene Ontology classification and KEGG enrichment analysis to highlight the function of the differentially expressed genes. The authors selected six pathways: cellular response to unfolded protein (GO:0034620), inflammatory response (GO:0006954), cytokine activity (GO:0005125), TNF signaling pathway(ko04668), influenza A (ko05164) and pathways in cancer(ko05200). The results for 27 genes of these pathways split them in one group of 15 upregulated genes and one group of 12 downregulated genes in GFP-ORF3-expressing cells compared to GFP-expressing cells. For eight of these genes, the differential expression was validated by qRT-PCR.

Main comments:

This study is interesting in that it could contribute to understand the role of the HEV ORF3 protein in infected cells. However, the highlighted differentially expressed genes were not validated in an infection context and the study remains descriptive. Therefore, the title should mention “Descriptive analysis of mRNA Transcriptome Profiles….” rather than “Comprehensive analysis of mRNA Transcriptome Profiles…”.

Introduction section

The text classically introduces the question but the chosen references not always appropriately support the assertions:

-ref 1 and 2 do not review the clinical and epidemiological settings of HEV infection

- HEV belongs to the Hepeviridae family and the ICTV classification should be indicated. HEV-4 has a broad tropism and is a zoonotic strain. The name “swine HEV” could be discussed. I suggested to mention HEV-4 and to give the reference of the strain. The reference of the first animal HEV strain, swine hepatitis E virus (SHEV), isolated and characterized in piglets in the United States is lacking.

- ref 6 to 9 do not describe the HEV particles and do not show transmission electron microscopy assessing the particle size. HEV is a virus particle without envelope (and not capsule).

- What does “infecting pigs with toxins in their mouths” means? If this means “by oral route”, so the oral infection of pigs has been successfully performed by Cao (PNAS 2017). Please correct.

- ref 17 shows the inactivation of HEV by heating at 56°C during one hour. Please correct.

Also, many works have focused on the HEV ORF3 protein. These should be mentionned in the introduction to highlight the interest of the study of the impact of ORF3 expression on the transcriptome.

Methods section

§  In the method section 2.2, the ORF3 sequence comes from a HEV-4 strain but the Genbank strain reference has to be mentioned. The determination of the adenovirus titer of the ADV4-GFP strain is not mentioned. Was the ratio of the infectious titer to ADV genome equivalents similar for ADV4-ORF3 and control ADV4-GFP? If not, a bias could have been introduced in the further interpretation of RNA expression differences. In the following paragraph, please indicate the moi (multiplicity of infection) used for transduction.

§  In the method section 2.3 and 2.4, the information concerning the RNA sequencing method are redundant. Paragraph 2.4 should begin with the first data analysis, that is, the Cutadapt filter.

§  In this section, the indicator of sequencing quality could be explained to help non-specialized readers: Q20, FPKM box-plots. In particular, this would help understanding the sentence “the Q20 reached 99.98%” in the result section 3.1. Does it mean that 99.98% of the sequences were of sufficiently high quality?

Results section

§  Figures are not all mentioned in the main text and do not appear in the consecutive order. There are two “Figure 1” and two “Figure 2” but no “Figure 3”. This should not be submitted and has to be corrected.

§  A main point of the results section is the presentation of the differentially expressed genes and transcripts. Why showing both? The different results between genes and transcripts in Figures 4 to 6 are not commented. How do the authors reconciliate them? If the two analyzes are justified, a Venn diagram to show the recovery between both could be helpful in Figure 4.

§  The authors chose six pathways with high correlation with ORF3 but some with high scores were ignored (C3HC4-type RING finger domain binding in GO analysis, legionellosis in KEGG analysis…). The authors analyzed the relative expression of the first five genes of these pathways. This should report 30 genes rather than 27. What’s about the three lacking?

§  The RT-QPCR was performed for the 27 genes but results were obtained for 8 of them. Could the failure of the other experiments be explained? Was it related to differences in RNa amounts?

Discussion section

The effect of ORF3 on the pathways evidenced by this study is very interesting. The discussion lacks some main articles relative to HEV and these pathways.

References section

Reference 1 is incomplete

Comments on the Quality of English Language

I don't know because I am not a native English speaker.

Author Response

Comments and Suggestions for Authors

Jiao et al focused on the role of the ORF3 protein of the Hepatitis E virus (HEV) in infected cells. To this end, a transcriptome analysis of HepG2 cells transduced with adenovirus strains expressing the green fluorescent protein (GFP) alone or GFP tagged-ORF3 was performed.

The first part of the article describes the validation of the transcriptome sequencing method. The quality scores and sequence recovery was sufficiently high to compare the different samples and perform a comparative analysis between cells expressing GFP and GFP-ORF3.

The authors found 217 differentially expressed genes (and 1635 transcripts) and used Gene Ontology classification and KEGG enrichment analysis to highlight the function of the differentially expressed genes. The authors selected six pathways: cellular response to unfolded protein (GO:0034620), inflammatory response (GO:0006954), cytokine activity (GO:0005125), TNF signaling pathway(ko04668), influenza A (ko05164) and pathways in cancer(ko05200). The results for 27 genes of these pathways split them in one group of 15 upregulated genes and one group of 12 downregulated genes in GFP-ORF3-expressing cells compared to GFP-expressing cells. For eight of these genes, the differential expression was validated by qRT-PCR.

Main comments:

This study is interesting in that it could contribute to understand the role of the HEV ORF3 protein in infected cells. However, the highlighted differentially expressed genes were not validated in an infection context and the study remains descriptive. Therefore, the title should mention “Descriptive analysis of mRNA Transcriptome Profiles….” rather than “Comprehensive analysis of mRNA Transcriptome Profiles…”.

Response: Thank you for your constructive suggestion. We have changed the title to “Descriptive Comparative Transcriptomic Analysis of Genotype IV SHEV ORF3-expressing HepG2 Cells ”.

Introduction section

The text classically introduces the question but the chosen references not always appropriately support the assertions:

-ref 1 and 2 do not review the clinical and epidemiological settings of HEV infection

- HEV belongs to the Hepeviridae family and the ICTV classification should be indicated. HEV-4 has a broad tropism and is a zoonotic strain. The name “swine HEV” could be discussed. I suggested to mention HEV-4 and to give the reference of the strain. The reference of the first animal HEV strain, swine hepatitis E virus (SHEV), isolated and characterized in piglets in the United States is lacking.

Response: Thank you for your valuable suggestion. We have rephrased and quoted according to your request. The references we have added are as follows:

  1. Ahmed R, Nasheri N. Animal reservoirs for hepatitis E virus within the Paslahepevirus genus. Vet Microbiol. 2023 Mar;278:109618.
  2.   Meng XJ, Purcell RH, Halbur PG, Lehman JR, Webb DM, Tsareva TS, et al. A novel virus in swine is closely related to the human hepatitis E virus. Proc Natl Acad Sci U S A. 1997 Sep 2;94(18):9860-5.
  3.   Li S, He Q, Yan L, Li M, Liang Z, Shu J, et al. Infectivity and pathogenicity of different hepatitis E virus genotypes/subtypes in rabbit model. Emerg Microbes Infect. 2020 Dec;9(1):2697-2705.
  4. Yang D, Jiang M, Jin M, Qiu ZG, Shen ZQ, Cui WH, et al. Seroprevalence and evolutionary dynamics of genotype 4 hepatitis E virus in Shandong Province, China. World J Gastroenterol. 2014 Jun 28;20(24):7955-63.

- ref 6 to 9 do not describe the HEV particles and do not show transmission electron microscopy assessing the particle size. HEV is a virus particle without envelope (and not capsule).

Response: Thank you for your review. We have re screened the relevant references describing HEV virus particles as follows:

  1. Wang B, Meng XJ. Structural and molecular biology of hepatitis E virus. Comput Struct Biotechnol J. 2021 Apr 7;19:1907-1916.
  2. Baha S, Zhang M, Behloul N, Liu Z, Wei W, Meng J. Efficient production and characterization of immunogenic HEV-PCV2 chimeric virus-like particles. Vet Microbiol. 2022 May;268:109410.
  3. Wei B, Li H, Cheng M, Yang Y, Liu B, Tian Y, et al. NLRP3 Inflammasome Activation Mediates Hepatitis E Virus-Induced Neuroinflammation. J Viral Hepat. 2024 Nov;31(11):729-738.
  4.  Kobayashi T, Takahashi M, Ohta S, Hoshino Y, Yamada K, Jirintai S, et al. Production and Characterization of Self-Assembled Virus-like Particles Comprising Capsid Proteins from Genotypes 3 and 4 Hepatitis E Virus (HEV) and Rabbit HEV Expressed in Escherichia coli. Viruses. 2024 Aug 31;16(9):1400.

- What does “infecting pigs with toxins in their mouths” means? If this means “by oral route”, so the oral infection of pigs has been successfully performed by Cao (PNAS 2017). Please correct.

Response: Thank you for your careful review. We have revised this sentence to “However, replicating HEV infection mainly occurs through oral routes (15)”

  1. Cao D, Cao QM, Subramaniam S, Yugo DM, Heffron CL, Rogers AJ, et al. Pig model mimicking chronic hepatitis E virus infection in immunocompromised patients to assess immune correlates during chronicity. Proc Natl Acad Sci U S A. 2017 Jul 3;114(27):6914-6923.

- ref 17 shows the inactivation of HEV by heating at 56°C during one hour. Please correct.

Response:Thanks a lot. We have corrected this sentence Heating at 56 ° C for a long time (1 hour) or at a higher temperature (>65 ° C) for a short time can successfully inactivate HEV virus.

Also, many works have focused on the HEV ORF3 protein. These should be mentionned in the introduction to highlight the interest of the study of the impact of ORF3 expression on the transcriptome.

Response:Thank you for your valuable suggestion. We have added the content of ORF3 protein in the introduction.The added parts are as follows:

“SHEV ORF3 is a regulatory protein that alters the activity of selected transcription factors and cytoplasmic signaling pathways (25). The ORF3 protein interacts with microtubules and various host cell proteins to regulate the host environment for HEV replication. Importantly, ORF3 protein can promote the budding of viral particles into polycystic bodies. Finally, the polycystic body fuses with the plasma membrane, and viral particles are released from liver cells into the bloodstream or degraded by bile acid salts in the bile duct (26).”

Methods section

  • In the method section 2.2, the ORF3 sequence comes from a HEV-4 strain but the Genbank strain reference has to be mentioned. The determination of the adenovirus titer of the ADV4-GFP strain is not mentioned. Was the ratio of the infectious titer to ADV genome equivalents similar for ADV4-ORF3 and control ADV4-GFP? If not, a bias could have been introduced in the further interpretation of RNA expression differences. In the following paragraph, please indicate the moi (multiplicity of infection) used for transduction.

Response: Thank you for your valuable suggestion. For specific details on the preparation and infection of recombinant adenovirus AD-ORF3 and AD-GFP, please refer to our published articles:

  1.   Jiao H, Shuai X, Luo Y, Zhou Z, Zhao Y, Li B, et al. Deep Insight Into Long Non-coding RNA and mRNA Transcriptome Profiling in HepG2 Cells Expressing Genotype IV Swine Hepatitis E Virus ORF3. Front Vet Sci. 2021 Apr 29;8:625609.

Following your suggestion, we have added the following MOI”HepG2 cells were seeded into a 12-well plate (1 × 106 cells per well), infected with ADV4-ORF3 and ADV4-NC at 12 h post seeding at a multiplicity of infection (MOI) of 5:1”

  • In the method section 2.3 and 2.4, the information concerning the RNA sequencing method are redundant. Paragraph 2.4 should begin with the first data analysis, that is, the Cutadapt filter.

Response: Thanks a lot. We have made the modifications of “Paragraph 2.4” according to your suggestion. In the method section 2.3, the information concerning the RNA sequencing method are deleted and cited the reference.

  1.  Kim D, Paggi JM, Park C, Bennett C, Salzberg SL. Graph-based genome alignment and genotyping with HISAT2 and HISAT-genotype. Nat Biotechnol. 2019 Aug;37(8):907-915.

  • In this section, the indicator of sequencing quality could be explained to help non-specialized readers: Q20, FPKM box-plots. In particular, this would help understanding the sentence “the Q20 reached 99.98%” in the result section 3.1. Does it mean that 99.98% of the sequences were of sufficiently high quality?

Response: The Q20 value refers to the error probability given to the identified bases during the base calling process of sequencing If the quality value is Q20, Q20 reached 99.98%, it means the accuracy rate is 99.98%.

Results section

  • Figures are not all mentioned in the main text and do not appear in the consecutive order. There are two “Figure 1” and two “Figure 2” but no “Figure 3”. This should not be submitted and has to be corrected.

Response: Thanks a lot. We have made the correction of Figures order.

  • A main point of the results section is the presentation of the differentially expressed genes and transcripts. Why showing both? The different results between genes and transcripts in Figures 4 to 6 are not commented. How do the authors reconciliate them? If the two analyzes are justified, a Venn diagram to show the recovery between both could be helpful in Figure 4.

Response: Because our transcriptomic results identified 217 significantly differentially expressed genes and 1635 significantly differentially expressed transcripts (│Log2FC│≥1 and q≤0.5), each gene may correspond to multiple transcripts. Therefore, we present both results.

  • The authors chose six pathways with high correlation with ORF3 but some with high scores were ignored (C3HC4-type RING finger domain binding in GO analysis, legionellosis in KEGG analysis…). The authors analyzed the relative expression of the first five genes of these pathways. This should report 30 genes rather than 27. What’s about the three lacking?

Response: Thank you for your attention. Our ultimate goal is still to discover some new targets closely related to ORF3 function through transcriptome sequencing. Therefore, we will closely focus on this goal for verification. The six pathways with high correlation of ORF3 were obtained, which were cellular response to unfolded protein(GO:0034620), inflammatory response(GO:0006954), cytokine activity(GO:0005125), TNF signaling pathway(ko04668), influenza A (ko05164) and pathways in cancer(ko05200). Among them, │Log2FC│≥1 and q≤0.5 were selected as the conditions, the pathways in cancer(ko05200) only has 2 differentially expressed genes. Therefore, there are only 27 significantly differentially expressed genes in total.

  • The RT-QPCR was performed for the 27 genes but results were obtained for 8 of them. Could the failure of the other experiments be explained? Was it related to differences in RNa amounts?

Response: The primers were designed for qRT-PCR verification. A total of 27 differential genes were verified, of which 15 genes were up-regulated and 12 genes were down-regulated. As shown in the figure 11, only 8 genes were validated correctly, while the remaining 19 did not match the sequencing results. There are false positives in high-throughput sequencing results, so we can only screen for truly differentially expressed genes through repeated validation. The genes that are validated incorrectly either have incorrect expression trends or levels.

Discussion section

The effect of ORF3 on the pathways evidenced by this study is very interesting. The discussion lacks some main articles relative to HEV and these pathways.

Response: Thank you for your valuable suggestion. The main articles relative to HEV and these pathways were added as follows in Disscusion:

  1.   Xu J, Wu F, Tian D, Wang J, Zheng Z, Xia N. Open reading frame 3 of genotype 1 hepatitis E virus inhibits nuclear factor-κappa B signaling induced by tumor necrosis factor-α in human A549 lung epithelial cells. PLoS One. 2014 Jun 24;9(6):e100787.
  2.  Qian Z, Li T, Xia Y, Cong C, Chen S, Zhang Y, et al. Genotype 4 Hepatitis E virus replicates in the placenta, causes severe histopathological damage, and vertically transmits to fetuses. J Infect. 2023 Jul;87(1):34-45.
  3.  Zhou X, Xu L, Wang W, Watashi K, Wang Y, Sprengers D, et al. Disparity of basal and therapeutically activated interferon signalling in constraining hepatitis E virus infection. J Viral Hepat. 2016 Apr;23(4):294-304.
  4.  Salam GD, Kumar A, Kar P, Aggarwal S, Husain A, Sharma S. Serum tumor necrosis factor-alpha level in hepatitis E virus-related acute viral hepatitis and fulminant hepatic failure in pregnant women. Hepatol Res. 2013 Aug;43(8):826-35.
  5.  Liang QL, Nie LB, Zou Y, Hou JL, Chen XQ, Bai MJ, et al. Serological evidence of hepatitis E virus and influenza A virus infection in farmed wild boars in China. Acta Trop. 2019 Apr;192:87-90.

References section

Reference 1 is incomplete

Response: Reference 1 has been replaced.

Reviewer 2 Report

Comments and Suggestions for Authors

Although I have only read the Abstract, Introduction and up to 2.2 of the Materials and Methods section of this manuscript, I already have difficulty understanding the English in this manuscript and cannot provide an adequate review. The authors should use an English language editing service and resubmit a revised version of manuscript.

Comments on the Quality of English Language

Although I have only read the Abstract, Introduction and up to 2.2 of the Materials and Methods section of this manuscript, I already have difficulty understanding the English in this manuscript and cannot provide an adequate review. The authors should use an English language editing service and resubmit a revised version of manuscript.

Author Response

Although I have only read the Abstract, Introduction and up to 2.2 of the Materials and Methods section of this manuscript, I already have difficulty understanding the English in this manuscript and cannot provide an adequate review. The authors should use an English language editing service and resubmit a revised version of manuscript.

Response: Thank you for your constructive suggestion. We have comprehensively revised the writing of the entire manuscript and invited experts who are native English speakers for language improvement.

Round 2

Reviewer 1 Report

Comments and Suggestions for Authors

 I thank the authors for taking my comments into account.

 There are still many problems of form that have to be corrected:

Introduction

The red sentence of the first paragraph is redundant with the two first one. Please correct.

The sentence “HEV belongs to the hepatitis virus family. Hepatitis E is significantly different from hepatitis B” does not seem necessary (why evoking HBV here ?)

In the sentence “HEV is a spherical virus particle without a capsule, with a diameter of 32-34 nm, the term “capsule” should be replaced by “envelop”. The reference particles size is 27-34 nm (https://ictv.global/report/chapter/hepeviridae/hepeviridae)

In the following sentences, the terms “however” and “therefore” should be inverted : “However, replicating HEV infection mainly oc-curs through oral routes (15). Therefore…”

“polycystic bodies” are commonly names “multivesicular bodies (MVB)”. Please correct.

Methods

In paragraph 2.3, the extraction and sequencing methods mentioned in the first version have been unfortunately erased but seem necessary although the addition of reference 34.

Paragraph 2.6 give information relative to the result section ( and should move there) but does not describe how the data were compared. Mention the used tool if necessary.

Results

3.1.1. The paragraph “The horizontal and vertical coordinates represent the samples, and the color shading indicates the magnitude of the correlation coefficient between the two samples. The closer to red (the closer the coefficient is to 1) indicates a greater correlation; The closer to white, the less correlation” should be the legend of Figure 1 and the figure number must be called up in the text after “there is a high correlation between samples”.

3.4. Maybe should this paragraph begin with “Finally” rather than “Firstly”

This paragraph lacks a conclusive sentence about the concordance or not of the two approach.

Discussion

It would be easier for the reader if you grouped the discussion arguments relative to innate immune response, then UPR, etc…

Author Response

Introduction

The red sentence of the first paragraph is redundant with the two first one. Please correct.

Response: Thank you for your valuable suggestion. We have already made modifications.

The sentence “HEV belongs to the hepatitis virus family. Hepatitis E is significantly different from hepatitis B” does not seem necessary (why evoking HBV here ?)

Response: Thank you for your suggestion. We have revised and removed the content of HBV from this sentence.

In the sentence “HEV is a spherical virus particle without a capsule, with a diameter of 32-34 nm, the term “capsule” should be replaced by “envelop”. The reference particles size is 27-34 nm (https://ictv.global/report/chapter/hepeviridae/hepeviridae)

Response: Thank you for your constructive review. We have made revisions based on your suggestions.

In the following sentences, the terms “however” and “therefore” should be inverted : “However, replicating HEV infection mainly oc-curs through oral routes (15). Therefore…”

Response: Thank you for your valuable suggestion. We have already inverted these two words of “however” and “therefore”.

“polycystic bodies” are commonly names “multivesicular bodies (MVB)”. Please correct.

Response: Thanks a lot. We have corrected “polycystic bodies”to “multivesicular bodies (MVB)”.

Methods

In paragraph 2.3, the extraction and sequencing methods mentioned in the first version have been unfortunately erased but seem necessary although the addition of reference 34.

Response: The extraction and sequencing methods have been described in detail in our previously published article- reference 34, so they have been directly deleted here.

Paragraph 2.6 give information relative to the result section ( and should move there) but does not describe how the data were compared. Mention the used tool if necessary.

Response: We have made the modifications according to your suggestion and changed this paragraph to ”HepG2 cells were infected with the recombinant adenovirus of ADV4-NC (control) and ADV4-ORF3 for 24 h, total RNA was extracted and depleted ribosomal RNA, Paired-end lncRNA sequencing was performed. We selected significantly differentially expressed genes (│Log2FC│≥1 and q≤0.5 ) from transcriptome and mRNA expression profiles of lncRNA sequencing results. Venn diagram reference text advanced pie plot was performed using the OmicStudio tools at https://www.omicstudio.cn/tool.”

Results

3.1.1. The paragraph “The horizontal and vertical coordinates represent the samples, and the color shading indicates the magnitude of the correlation coefficient between the two samples. The closer to red (the closer the coefficient is to 1) indicates a greater correlation; The closer to white, the less correlation” should be the legend of Figure 1 and the figure number must be called up in the text after “there is a high correlation between samples”.

Response: Thank you for your careful review. We have revised it.

3.4. Maybe should this paragraph begin with “Finally” rather than “Firstly”

This paragraph lacks a conclusive sentence about the concordance or not of the two approach.

Response: Thanks a lot. We have already made modifications of this paragraph begin with “Finally” rather than “Firstly”. And added a concluding sentence

Discussion

It would be easier for the reader if you grouped the discussion arguments relative to innate immune response, then UPR, etc…

Response: We have made adjustments to the relevant content based on your suggestion for discussion.